# Investigating experiences of stigma and its impact on substance use recovery among residents of recovery residences in the United States: A scoping review protocol. BSGH 020

Gamji Rabiu Abu-Ba'are[1,2,3]☉, Sahil Hogarty[1,4]☉, Osman Wumpini Shamrock●[1]*, Holly Russell[5], Kate Manchisi[6], Van Smith[6], Amy Mericle[7]

1 Behavioral, Sexual, and Global Health Lab, School of Nursing, University of Rochester, Rochester, New York, United States of America, 2 Department of Public Health Sciences, University of Rochester, Rochester, New York, United States of America, 3 Center for Interdisciplinary Research on AIDS, School of Public Health, Yale University, New Haven, CT, United States of America, 4 Davidson College, Davidson, North Carolina, United States of America, 5 University of Rochester Medical Center, Rochester, New York, United States of America, 6 Recovery Houses of Rochester, Rochester, New York, United States of America, 7 Alcohol Research Group, Emeryville, California, United States of America

☉ These authors contributed equally to this work.
* osmanwumpini_shamrock@urmc.rochester.edu

## Abstract

### Objective

The objective of this scoping review is to systematically review the literature on stigma experienced by residents in recovery residences and its impact on substance use recovery outcomes.

### Method

The review will use the PRISMA-ScR framework to identify studies focused on stigma and recovery in recovery residences published in English within the United States since 2000, including qualitative, quantitative, and mixed-methods studies. Data will be extracted and analyzed thematically to identify gaps in the literature and inform future research and policy development.

### Result

Preliminary findings suggest that stigma, including labeling and discrimination, significantly hinders recovery by promoting secrecy and withdrawal among residents. Proximity to recovery residences has been shown to reduce community stigma, indicating the potential for better integration and acceptance.

### Conclusion

This study aims to provide a comprehensive understanding of stigma in recovery residences, its effects on substance use recovery, and recommendations for creating

**Data Availability Statement:** No datasets were generated or analysed during the current study. All relevant data from this study will be made available upon study completion.

**Funding:** The author(s) received no specific funding for this work.

**Competing interests:** The authors have declared that no competing interests exist.

supportive recovery environments. The significance of this study lies in its potential to inform policy, practice, and research, highlighting the need for stigma reduction to improve recovery outcomes in recovery residences. By addressing gaps in the literature, the findings will contribute to developing more effective interventions and supportive environments for individuals recovering from substance use.

## Introduction

According to the National Center for Health Statistics at the Center for Disease Control, nearly 108,000 people in the U.S. died from drug-involved overdose in 2022 from illicit or prescription drugs [1]. The use of synthetic opioids other than the use of methadone (primarily fentanyl) has been the driving force of drug overdose deaths, with a staggering 7.5-fold increase from 2015 to 2022 [1]. Not only are many lost due to illicit drug use, but the economic cost of drug abuse is estimated to be about 193 billion dollars annually in the United States, encompassing healthcare, lost productivity, and criminal justice costs [2]. Despite these well-known consequences of substance use (SU) [3–5] and its effects on physical health, psychological health, and quality of life, only a tiny fraction of people receive any treatment [2]. Presently, only an estimated 10% of individuals in America with alcohol and drug use disorders that meet the criteria for a diagnosable substance use disorder (SUD) receive any form of specialized treatment, indicating a staggeringly small number of individuals seeking help [2, 6]. SU affects physical and emotional well-being, familial and other relationships, education and career attainment, financial and criminal involvement, and spiritual health [2, 7–10].

Recovery residences (RR), also known by various names such as recovery homes, sober homes, halfway houses, etc., have emerged as a critical component in the continuum of care for SUD. The National Alliance for Recovery Residences has defined RR as "safe, healthy, and substance-free living environments that support individuals in recovery from addiction (varying widely in structure), all centered on peer support and a connection to services that promote long-term recovery" [11–13]. RR significantly differs from traditional biomedical treatment models by emphasizing the social model of recovery, which highlights the importance of experiential knowledge and peer interaction [12]. Research has consistently demonstrated positive outcomes among individuals entering RR [14–16]. Residents who enter RR "sustained reductions in substance use and legal problems and an increase in employment over 18 months," those improvements were maintained over 18 months even long after the residents had left their RR [17, 18]. For example, recovery services in RR had seen 6-month abstinence rates improve from 11% at baseline to 68% in 6 to 12 months [19].

Additionally, studies have found that patients transitioning from biomedical care facilities to recovery housing experienced "longer stays in outpatient treatment," underscoring the role of peer-led environments in sustaining recovery [14, 20]. The 'helper-therapy' principle, where peers benefit from receiving and providing support, is a cornerstone of these recovery communities, contributing to improved mental health and life satisfaction among residents [12]. Peer-based harm reduction initiatives, such as syringe exchange programs and overdose prevention sites in Vancouver, have further illustrated the positive impact peer-led interventions have on mental health status and improvement in life satisfaction rates [21]. These programs allow peer interaction to diffuse the tension and stigma associated with SU [22]. However, despite their successes, RR faces significant challenges rooted in stigma and prejudice. Services delivered by peers in these environments can be viewed as inferior compared to those provided in

traditional clinical settings, which may contribute to ongoing negative attitudes from both the public and professionals [12].

Stigma is a broad term for individual or group differences associated with negative stereotypes and behaviors [23, 24]. Labels such as "drug addict" and "alcoholic" are highly stigmatized and evoke negative responses from the community [24, 25]. Stigma manifests in various forms, including labeling, discrimination, devaluation, and internalization, and can occur at individual, community, and societal levels [26, 27]. Individuals with "stigmatized markers," such as those who use illicit drugs, often feel isolated, morally and criminally policed and may be less inclined to disclose their status and seek support in RR [28]. Furthermore, it was reported that residents of R.H.'s or individuals receiving treatment for SU experience high levels of enacted, perceived, and self-stigma [26, 27]. Current treatment systems may inadvertently stigmatize people in recovery, as those with more prior episodes of treatment reported higher frequencies of stigma-related rejection, even after controlling for current functioning and demographic variables [26, 27]. Due to stigmatization, residents of these houses may resort to coping mechanisms like secrecy and withdrawal from their communities [29]. There are specific experiences of enacted stigma among RR residents, including believing that people mistreated them because of their SU (60%), feeling that others were afraid of them (46%), sensing that some family members gave up on them (45%), and experiencing rejection from friends (38%) [27, 29].

Culturally, SU is still primarily regarded as an "immoral or inept lifestyle choice for which affected individuals are fully culpable," reinforcing the stigma that individuals with SUD are defective characters rather than recognizing SU as a chronic and potentially fatal health condition [29–31]. This misconception ensures that SU retains its potency as a sign of a defective character, leading to unfair treatment and social ostracization of residents in RR [29]. However, studies show that physical proximity can reduce opposition to RR and their residents [29, 32]. For example, neighbors of group homes reported fewer perceived threats to personal safety and property values than those without such proximity [29, 33]. Similarly, community residents living next door to Oxford Houses in Northern Illinois had more favorable attitudes toward them than those living a block away [29, 32]. These findings suggest that integrating RR residents with the public can help dilute stigma by demonstrating that residents are ordinary citizens striving to better themselves [29].

Despite existing research on stigma and recovery housing [29], a comprehensive evaluation of the literature that explores the various types of stigmas faced by residents of recovery housing or how stigma affects their paths to recovery is still lacking. Moreover, stigma on residents, when studied, is mostly disjointed; hence, studies that bring together these experiences in a coherent manner may be more informative on developing sustainable paths to recovery among residents. Understanding the stigma experienced by residents in R.H. is crucial for multiple stakeholders and policies around SU. Knowledge of stigma among residents can inform the development of more supportive policies to enhance recovery outcomes. It will offer practitioners insights into creating more effective and inclusive intervention strategies. For researchers, it highlights existing gaps and sets the stage for future studies. By addressing stigma, R.H. will present a more welcoming and practical approach to improving SU outcomes for residents. Therefore, the significance of this scoping review lies in the fact that no existing study has been done to address the stigma people face in RR.

## Methods and analysis

We will use the Preferred Reporting Items for Systematic Reviews and Meta-Analyses Extension for Scoping Reviews (PRISMA-ScR) by [34] to conduct and report this scoping review.

To conduct a scoping review, it remains imperative to develop a protocol to clarify the purpose and methodology of the evaluation. Thus, ensuring transparency in the scoping review process will prevent the duplication of efforts around stigma and recovery among recovery home residents.

We aim to employ statistical analysis as a component of our methodology, utilizing descriptive statistics to summarize stigma frequency and inferential methods to examine associations between stigma experiences and recovery outcomes. This approach is designed to ensure that the study produces reliable and meaningful data trends, contributing both to academic knowledge and practical implications for recovery residences.

## Ethics and considerations

The study will not require review by an ethical review board as it does not involve human subjects or secondary data from human subjects. We will publish the scoping review findings in a peer-reviewed journal for other researchers, practitioners, and stakeholders in the field of SU to access and use to inform their research or practice. We will also present the findings at relevant conferences and events on substance recovery to inform a wider audience of the state of science around stigma and recovery concerning recovery house residents. Our findings will identify gaps in the literature and inform future studies and interventions to address stigma in the context of RR.

## Patient and public involvement

The scoping review will focus on existing published studies on stigma and its impact on recovery among residents of RR. Hence, it will not involve human participants or data collection.

**Inclusion criteria.** The scoping review will include articles focusing on stigma and recovery residents only. To ensure relevance to current circumstances, the studies must be conducted within the United States, in English, and published on or after 2000.

**Exclusion criteria.** To ensure relevance to current circumstances within recovery homes, we will exclude studies conducted outside the United States and published after 2000. We will also exclude housing literature that is not for individuals in recovery from SU. We will also exclude review articles and reports that do not include primary or secondary data collected from recovery homes.

**Types of studies.** We will review all studies that analyzed data, such as quantitative, qualitative, and mixed methods data. We will also review experimental and observational studies regardless of the methodological approaches. We will, however, exclude literature-based papers.

## Search strategy

**Identifying sources.** Based on our standard practice and comparable to what was done in similar protocols [35, 36], a librarian from the University of Rochester Medical Center, who will be co-authors in this study, will create the search strategy for the scoping review. Other co-authors will review the search strategy and provide insights on areas for improvement and ways to identify articles that may not be readily available in library databases. To ensure the most recent and up to data literature reflects our study's focus, all article searches will not be done earlier than one year before the time of submission.

**Electronic database searching.** The librarian will conduct a literature search in databases such as PubMed (NCBI), Web of Science Core Collection (Clarivate), Embase (Elsevier), and Scopus (Elsevier). The librarian will search using keywords encompassing the commonly used terms relevant to the research topics. The keywords will include alcohol, alcohol rehab, halfway

houses, therapeutic communities' recovery residences, recovery houses, sober living homes, recovery group houses, Oxford Houses, stigma, recovery residences, SU recovery, and drug use recovery.

**Grey literature searching.**  We plan to search for grey literature in databases such as the Global Index Medicus (WHO), MedNar (Deep Web Technologies), and Central Registry of Controlled Trials (Cochrane).

**Data screening.**  Using the inclusion and exclusion criteria, the librarian will export all identified articles to Covidence and remove duplicate articles. We will then subject the articles to a two-step screening process: 1) abstract and title screening and 2) full-text screening). We will divide the articles and assign each article based on availability to two reviewers to conduct the first phase of screening the title and abstract. We will then assign the remaining articles to two reviewers to review for inclusion or exclusion in the data extraction stage. At the second screening stage, the two reviewers assigned to each article will meet to resolve any conflicts; a third reviewer will be invited to provide additional inputs should the two reviewers fail to reach a consensus on including or excluding the article based on the criteria.

### Data extraction

**Content.**  Using Covidence, we will assign each article in the screening stage to two reviewers to extract relevant information. We will create a data extraction questionnaire that reviewers will use to conduct the data extraction. The questionnaire will ask for essential details such as the study title, author, year of publication, study design, aim, and date. We will also ask for a description of the study participants, methods of recruitment, and number of participants, as well as key findings from the study and recommendations.

**Process.**  We will conduct a pilot test of screening and data extraction of selected articles among all reviewers to ensure consistency and understanding of the use of the screening and extraction tools in evidence and on the extraction form. We will then provide necessary clarifications to the team and modify the tools as needed.

**Analysis and reporting.**  This study's findings will follow the PRISMA-ScR guidelines [34], similar to what have been used in previous studies [35, 37]. After data extraction, two reviewers assigned to each article will meet to consolidate their findings into a single document per article. Then, two authors will independently perform a thematic analysis [35, 36] on each document. They will subsequently compare their analyses to generate a unified thematic table covering all articles that have completed the extraction process. While we will not combine quantitative data as in systematic reviews, we will summarize the articles to present the number of studies, topics of inquiry, and descriptive findings. Finally, a third author will review and compile these summaries for inclusion in the final manuscript intended for publication in a scientific journal.

**Outcome.**  The primary outcome of the scoping review will encompass a broad characterization of the experiences of stigma among residents of recovery homes. It will also provide information on the current state of science on stigma in RR and gaps in the literature that need to be filled. The secondary outcome of the scoping review will encompass the impact of stigma experienced by residents of RR and how such stigma impacts their SU recovery outcomes. These findings can inform practice and research into improving SU recovery outcomes among residents in the United States.

## Discussion

Despite efforts to decrease SU and promote recovery, it persists due to personal and social challenges such as stigma. RR offers a socio-ecological intervention to address SU. This is

salient given that only 6.8% of those who need S.U. treatment received it at a specialized facility. Recovery housing differs from the traditional clinical recovery process by using a social model [12, 38–40] that houses residents to support recovery efforts via peer mentoring, promoting autonomy, and creating a culturally safe space for men of color. However, stigma has been identified as a factor affecting recovery homes and has received recent attention in research and practice. Nonetheless, the extent to which studies have examined stigma experienced among recovery residents remains unknown. Therefore, we aim to use this scoping review to establish the state of science, including intervention among recovery residents to address stigma and to improve SU outcomes among residents.

We also intend to present our findings in a manner that highlights statistically significant patterns and associations within the data, such as the relationship between stigma types and withdrawal behaviors. These associations offer valuable insights into stigma experiences. Our findings will explain existing gaps in studies and interventions and inform future directions of studies and interventions to help improve SU outcomes among residents of recovery homes.

## Strengths and limitations of the study

1. This scoping review is strengthened by an experienced medical librarian who has designed a search strategy to ensure the comprehensive inclusion of peer-reviewed articles from multiple databases relevant to SU research.

2. By taking an interdisciplinary approach, we hope this scoping review will offer statistical insights that contribute to a holistic understanding of stigma in recovery residences. We anticipate that integrating perspectives from clinical research would further enrich the applicability of these findings

3. The scoping review will also provide an update on the current state of science on stigma as a critical factor in SU prevention and treatment.

4. The focus on RR, which continues to grow across the United States, will ensure that the findings provide relevant information to inform practice and research in using RR to address stigma and improve SU outcomes.

5. The focus on the United States will limit its relevance to the context of the United States and thus cannot be used as a basis for practice or research elsewhere. It can, however, inform the design and implementation of studies within the context of the United States and, to some extent, a replica in other areas when adapted.

## Author Contributions

**Conceptualization:** Gamji Rabiu Abu-Ba'are, Sahil Hogarty, Osman Wumpini Shamrock.

**Formal analysis:** Gamji Rabiu Abu-Ba'are.

**Investigation:** Gamji Rabiu Abu-Ba'are, Sahil Hogarty, Osman Wumpini Shamrock.

**Methodology:** Gamji Rabiu Abu-Ba'are, Sahil Hogarty, Osman Wumpini Shamrock.

**Supervision:** Gamji Rabiu Abu-Ba'are, Osman Wumpini Shamrock, Amy Mericle.

**Visualization:** Gamji Rabiu Abu-Ba'are, Sahil Hogarty, Osman Wumpini Shamrock.

**Writing – original draft:** Gamji Rabiu Abu-Ba'are, Sahil Hogarty, Osman Wumpini Shamrock, Holly Russell, Kate Manchisi, Van Smith, Amy Mericle.

**Writing – review & editing:** Gamji Rabiu Abu-Ba'are, Sahil Hogarty, Osman Wumpini Shamrock, Holly Russell, Kate Manchisi, Van Smith.

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
