## [Decision Letter · Decision Letter 0]

31 Oct 2024

PONE-D-24-36007Investigating experiences of stigma and its impact on substance use recovery among residents of recovery residences in the United States: A scoping review protocol. BSGH020PLOS ONE

Dear Dr. Shamrock,

Thank you for submitting your manuscript to PLOS ONE. After careful consideration, we feel that it has merit but does not fully meet PLOS ONE’s publication criteria as it currently stands. Therefore, we invite you to submit a revised version of the manuscript that addresses the points raised during the review process.

We look forward to receiving your revised manuscript.

Kind regards,

Ram A Sharma, MD

Academic Editor

PLOS ONE

Journal Requirements:

-https://leg.colorado.gov/sites/default/files/images/recovery_housing_issue_brief_may_2017_4.pdf

(among others)

In your revision ensure you cite all your sources (including your own works), and quote or rephrase any duplicated text outside the methods section. Further consideration is dependent on these concerns being addressed.

Additional Editor Comments:

The reviewers have made some valid comments. I believe it is necessary to address them, they are minor revisions but adds to the thought process behind the manuscript and answers some of the questions that reviewers posed. Overall, I appreciate the authors effort that they have put into this manuscript.

Reviewers' comments:

Reviewer's Responses to Questions

**Comments to the Author**

1. Does the manuscript provide a valid rationale for the proposed study, with clearly identified and justified research questions?

Reviewer #1: Partly

Reviewer #2: Yes

2. Is the protocol technically sound and planned in a manner that will lead to a meaningful outcome and allow testing the stated hypotheses?

Reviewer #1: Partly

Reviewer #2: Yes

3. Is the methodology feasible and described in sufficient detail to allow the work to be replicable?

Reviewer #1: Yes

Reviewer #2: Yes

4. Have the authors described where all data underlying the findings will be made available when the study is complete?

Reviewer #1: No

Reviewer #2: Yes

5. Is the manuscript presented in an intelligible fashion and written in standard English?

Reviewer #1: Yes

Reviewer #2: Yes

6. Review Comments to the Author

You may also provide optional suggestions and comments to authors that they might find helpful in planning their study.

Reviewer #1: This is an interesting and important topic to be studied.

I have some minor queries and suggestions-

1) Introduction section will be better if it is shorter and more concise.

2) Why you are choosing grey literature? How you can ensure scientific authenticity, like you have mentioned, peer reviewed contents?

3) Stigmas are already identified. What will be the new or any extra findings as per your expectations? I mean, why your review will be different and new?

4) Why you are planning to use thematic analysis approach? What components of validity in qualitative research will be addressed?

Reviewer #2: Important note: This review pertains only to ‘statistical aspects’ of the study and so ‘clinical aspects’ [like medical importance, relevance of the study, ‘clinical significance and implication(s)’ of the whole study, etc.] are to be evaluated [should be assessed] separately/independently. Further please note that any ‘statistical review’ is generally done under the assumption that study specific methodological [as well as execution] issues are perfectly taken care of by the investigator(s). This review is not an exception to that and so does not cover clinical aspects {however, seldom comments are made only if those issues are intimately / scientifically related & intermingle with ‘statistical aspects’ of the study}. Agreed that ‘statistical methods’ are used as just tools here, however, they are vital part of methodology [and so should be given due importance]. I look at the manuscript in/with statistical view point, other reviewer(s) look(s) at it with different angle so that in totality the review is very comprehensive. However, there should be efforts from authors side to improve (may be by taking clues from reviewer’s comments). Therefore, please do not limit the revision only (with respect) to comments made here.

COMMENTS: Though the study/manuscript is rather (almost) excellent, I have little/minor different opinion / observations/concerns or rather questions regarding (very) few issues (only two) which are given below:

Firstly, I noted that your ABSTRACT is very well drafted (in my opinion), but is ‘assay type’. It is preferable to divide the ABSTRACT with small sections like ‘Objective(s)’, ‘Methods’, ‘Results’, ‘Conclusions’, etc. which is an accepted practice of most of the good/standard journals [including this one, though ‘The PLoS One Guidelines to Authors’ did not specify an Abstract format, it is desirable]. It will definitely be more informative then, I guess, whatever the article type may be {though Section headings may differ for different Article Types [example: Study Protocol (for systematic review)]}.

Secondly (only a suggestion, may just consider, need not follow), I guess the title could be “The protocol of scoping* review on: Investigating experiences of stigma and its impact on substance use recovery among residents of recovery residences in the United States - BSGH020” instead of present “Investigating experiences of stigma and its impact on substance use recovery among residents of recovery residences in the United States: A scoping review protocol. BSGH020”.

*[A scoping review is a systematic method for identifying and mapping the available evidence on a topic, concept, or issue]

Will you please add ‘What is BSGH020 and why included it in the title. Purpose of publication of protocol of ‘A scoping review’ is also expected to be mentioned.

As pointed out in ‘important note’ above “This review pertains only to ‘statistical aspects’ of the study and so ‘clinical aspects’ should be assessed separately/independently. In my opinion, to make this article more acceptable (which is already of acceptable level), a small amount of re-vision may be needed.

7. PLOS authors have the option to publish the peer review history of their article (what does this mean?). If published, this will include your full peer review and any attached files.

Reviewer #1: **Yes: **Panchanan Acharjee

Reviewer #2: **Yes: **Dr. Sanjeev Sarmukaddam

---

## [Author Response · Author response to Decision Letter 0]

4 Nov 2024

Dear reviewer,

We have addressed all the recommendations provided by the reviewer. However, we would like to retain the title as it is: “Investigating experiences of stigma and its impact on substance use recovery among residents of recovery residences in the United States: A scoping review protocol. BSGH 020.” The title BSGH is the acronym for Behavioral, Sexual and Global Health Lab. Our manuscripts are abbreviated with the first letters of the lab title and numbers to keep track of our publications. 

We have independently addressed the statistical aspects separate from the clinical aspects in this protocol and highlighted these sections within the study document in yellow.

We hope these changes meet your recommendations and look forward to a positive response soon.

Sincerely 

Osman Wumpini Shamrock PhD, MPhil 

School of Nursing, University of Rochester 

225 Crittenden Blvd, Rochester, NY, 14642 

+1 (607) 761 8189

---

## [Decision Letter · Decision Letter 1]

14 Nov 2024

Investigating experiences of stigma and its impact on substance use recovery among residents of recovery residences in the United States: A scoping review protocol. BSGH020

PONE-D-24-36007R1

Dear Dr. Shamrock,

We’re pleased to inform you that your manuscript has been judged scientifically suitable for publication and will be formally accepted for publication once it meets all outstanding technical requirements.

Kind regards,

Ram A Sharma, MD

Academic Editor

PLOS ONE

Additional Editor Comments (optional):

Reviewers' comments:

Reviewer's Responses to Questions

**Comments to the Author**

1. Does the manuscript provide a valid rationale for the proposed study, with clearly identified and justified research questions?

Reviewer #2: Yes

2. Is the protocol technically sound and planned in a manner that will lead to a meaningful outcome and allow testing the stated hypotheses?

Reviewer #2: Yes

3. Is the methodology feasible and described in sufficient detail to allow the work to be replicable?

Reviewer #2: Yes

4. Have the authors described where all data underlying the findings will be made available when the study is complete?

Reviewer #2: Yes

5. Is the manuscript presented in an intelligible fashion and written in standard English?

Reviewer #2: Yes

6. Review Comments to the Author

You may also provide optional suggestions and comments to authors that they might find helpful in planning their study.

Reviewer #2: COMMENTS: All most all the comments made on earlier draft are/were considered positively & are attended satisfactorily. I recommend the acceptance.

However, please note that although you have independently addressed the statistical aspects, the document/letter “Response to Reviewer” is/was not in desired [comment-response] format (therefore the re-review was difficult).

7. PLOS authors have the option to publish the peer review history of their article (what does this mean?). If published, this will include your full peer review and any attached files.

Reviewer #2: **Yes: **Dr. Sanjeev Sarmukaddam

---

## [Editor Report · Acceptance letter]

19 Nov 2024

PONE-D-24-36007R1 

PLOS ONE

Dear Dr. Shamrock, 

I'm pleased to inform you that your manuscript has been deemed suitable for publication in PLOS ONE. Congratulations! Your manuscript is now being handed over to our production team.

Kind regards, 

on behalf of

Dr. Ram A Sharma 

Academic Editor

PLOS ONE